# Filter redistribution templates for iteration-less convolutional model reduction

## ABSTRACT

Automatic neural network discovery methods face an enormous challenge caused by the size of the search space. A common practice is to split this space at different levels and to explore only a fraction of it. On one hand, neural architecture search methods look at how to combine a subset of layers to create an architecture while keeping a predefined number of filters in each layer. On the other hand, pruning techniques take a well known architecture and look for the appropriate number of filters per layer. In both cases, the exploration is made iteratively, training models several times during the search. Inspired by the constraints and advantages of these two approaches, we propose a straight-forward and fast option to find models with improved characteristics. We apply a small set of templates, that have been heuristically and experimentally evaluated, to make a one-shot redistribution of the number of filters in an already existing neural network. When compared to the initial base models we found that the resulting architectures, when trained from scratch, surpass the original accuracy even after been reduced to fit to the original amount of resources. Specifically, we show accuracy improvement for some network-task pairs of up to 5.5%, a reduction of up to 45% in parameters and 60% reduction in memory footprint.

## 1 INTRODUCTION

Convolutional neural networks are built by stacking layers of neurons following the principle explained by Fukushima's Neocognitron model (Fukushima, 1980). The Neocognitron design made neural networks invariant to shift in feature locations by arranging cells locally connected in a hierarchical architecture.

Instead of connecting every neuron of the previous layer to all the neurons in the next layer, convolutional networks connections are only made to a small region. Given that different regions share the same weights, the layer can be implemented as a convolution operation using the set of shared weights known as kernel. The complete input is processed by shifting the kernel at uniform steps, normally overlapping parts of the input. To improve shift invariance, convolutional networks needs to rely less in the exact position of a feature and a simple solution is to have a lower resolution performed by averaging the values of neighbouring points in the image in a operation known as spatial subsampling.

An important consideration to create a convolutional network model is the number of filters, required at every layer. The Neocognitron implementation for example, keeps a fixed number of filters for each layer in the model. A very common practice has been to use a by-pyramidal architecture. The number of filters across the different layers is usually increased as the size of the feature maps decrease. This pattern was first proposed by LeCun et al. (1998) with the introduction of LeNet and can be observed in a diverse set of models such as VGG, ResNet and MobileNet (See Figure 5 in Appendix). Even models obtained from automatic model discovery, like NASNet, follow this principle inasmuch as neural network methods are mainly formulated to search for layers and connections.

It can be found in (LeCun et al., 1998) that the reason behind this progressive increase in the number of kernels is to compensated a possible loss of the representation caused by the spatial resolution reduction. In recent models, what seems to be the real reason is a practical issue (Chu & Krzyżak, 2014), to improve performance by keeping a constant number of operations in each layer.

The Pyramidal distribution of filters has perpetuated over two areas of model discovery. The methods in Automatic Neural Architecture Search (Liu et al., 2018a; Tan et al., 2019) explore the models built by combining a predefined set of layers, commonly with a pyramidal distribution of filters. On the other side, Network Pruning aims to reduce a model computational resources demands by selecting and removing weights that match some rule, commonly the closest to zero values, but starting from models that present this pyramidal distribution.

To the best of our knowledge, it remains unknown if this pyramidal distribution of filters is also beneficial to different aspects of model performances other than the number of operations. How the distribution of resources in a deep network model affect accuracy, memory footprint, inference time and model compression level are also of high interest in a multi-performing space that networks have to operate in.

This paper explores the topic by comparing models against versions of themselves with the same general structures but with the distribution of filters across the layers changed. We present a straightforward and fast to implement method for model discovery, that takes the goal of structured pruning methods, that is finding the lowest number of filters for a model while maintaining accuracy. Our exploration technique only tests over a small subset of diverse filter distributions, that we call templates, and that reduce the model proportionally to match some resource budget. Our experiments show that by using our proposed templates, the resulting models keep comparable accuracy as the original models in classification tasks but present reductions in the number of parameters and/or memory footprint.

The contributions of this paper are the following: 1) it provides evidence that pyramidal distribution of filters in convolutional network models is usually optimised for a distributed GPU operation across layers, and simple changes to that distribution leads to improvements in metrics such as number of parameters or memory footprint; 2) it highlights that most recent models, which have had a more detailed tuning in the filter distribution, present resiliency in accuracy to changes in the filter distribution, a phenomena that requires further research and explanation; 3) it shows that redistributing filters in a model and then applying a width multiplier operation can be seen as a pruning technique which produces smaller models than just applying the width multiplier to the original models; 4) it gives classical models a repositioning of their merit when measuring other practical implementation resources including and beyond the number of operations. And, 5) A practical and fast alternative approach for model search that can work in addition of conventional architecture search and network pruning.

The rest of the paper is structured as follows: Chapter 2 explores the most recent methods to reduce the size of neural network architectures. Chapter 3 describes the set of templates for filter distribution and how to implement this change in a convolutional network model. Chapter 5.1 compares dissimilar allocations of filters and their effect in model performances. Finally, chapter 5 briefly explains the findings derived from experiments.

## 2 RELATED WORK

The process of designing a Neural Networks is a task that largely has been based on experience and experimentation that consumes a lot of time and computational resources. Of note are reference models such as VGG, ResNet, Inception and similar that have been developed entirely or with significant use of heuristics. With the increase in the use of Neural Networks, and particularly Convolutional Networks for computer vision problems, a mechanism to automatically find the best architecture has become a requirement in the field of Deep Learning. Although some works have been published several years ago (Pinto et al., 2009; Kuri-Morales, 2014) trying to address the topic of automatic architecture generation, they have not provided competitive results compared to many hand-crafted architectures. However, current works start to lead the state of the art models (Elsken et al., 2018).

But even with automatic methods, one key feature that constantly has been adopted is the selection of the number of filters in each layer in the final model. The filters are set in such a way to have an increasing number as the layers go deeper. Pruning methods have done some work in this field

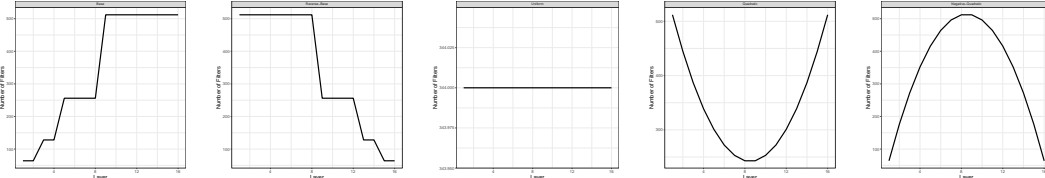

Figure 1: Filters per layer using the proposed templates for filter redistribution in the VGG 19 model. Base distribution, which is the original distribution, shows the common design of growing the filters when resolution of feature maps decreases in deeper layers.

but with the belief that the weights obtained at the end of the training process are important to the pruning method.

One common characteristic of many model discovery methods is that the search process is very time consuming, normally in the order of thousands of GPU hours. In this sense, a remarkable improvement is presented by Liu et al. (2018b). That method uses a relaxation condition to transform the selection of layers in the architecture to a continuous space using a *softmax* function. The relaxation allows to perform simultaneous search of weights and architecture using gradient descent. The method converges after one day of GPU time.

On the side of Pruning methods the search also involves training models for several iterations to select the correct weights to remove (Frankle & Carbin, 2018; He et al., 2019), or at least increasing the computation during the training when doing jointly training and search (Leclerc et al., 2018). Recently, Liu et al. (2018c) suggest that accuracy obtained by pruning techniques can be reached by training from scratch.

Our work relates to (Gordon et al., 2018) in the sense that their method is not restricted to reducing filters but also to increase them to see if the increment is beneficial. Our approach differs however, because it does not require to train the model other than in the final stage, after making some predefined changes to the number of filters using the redistribution template.

## 3  FILTER DISTRIBUTION TEMPLATES

Recent pruning methods such as Gordon et al. (2018); Leclerc et al. (2018), have shown different filter distribution patterns emerging when reducing models like VGG that defy the notion of pyramidal design as the best distribution for a model. This is a motivational insight into what other distributions can and should be considered when designing models. On one side the combinatorial space of distributions make this a challenging exploration, on the other however, it importantly highlights the need to pursue such exploration if gains in accuracy and overall performance can be made. In this work, rather than attempting to find the optimal filter distribution with expensive automatic pruning or growing techniques, we propose to first adjust the filters of a convolutional network model via a small number of pre-defined templates. These templates such as those depicted in figure 1, are inspired by existing models that have already been found to perform well and thus candidates that could be beneficial for model performance improvement beyond the number of operations. Performance criteria such as accuracy, memory footprint and inference time and model compression level are arguably as important as the number of operations required.

In particular, we adopt as one template a distribution with a fixed number of filters, as with the original Neocognitron design, but also other templates inspired by the patterns found in (Gordon et al., 2018) where at least three behaviours are present in different blocks from the resulting ResNet101 model: 1) filters increase in deeper layers, 2) filters agglomerate in the centre and 3) filters are reduced in the centre of the block. (Leclerc et al., 2018) shows also a filter pattern with more filters in the centre of a VGG model. Based on these observations we define the templates we use in this work.

We define a convolutional neural network base model as a set of layers $L = 1, ..., D + 1$ each of them containing a number of filters $f_l$ and a total number of filters $F = \sum_{l=1}^{D} f_l$. We want to test

if the common heuristic of distributing $F$ having $f_{l+1} = 2f_l$ each time the feature map is halved, is advantageous to the model over other distributions of $f_l$ when evaluating performance, memory footprint and inference time.

It should be noted that that final layer $D + 1$ remains with the same number of filters according to the task under evaluation, therefore it is not taken into account in the equations. Another important consideration is that, in architectures composed for modules or blocks (ResNet and Inception), it is easier to change the number of filters in the module as a whole than to change the filters in each particular layer inside the module and then to ensure that concatenations and additions from the previous layers match the correct number of filters. We have adopted then, this assumption of taking blocks as single layers to set the values of $f_l$. As a result, a final ResNet or Inception module marked with $f_l$ filters, is set to $f_l$ filters in each layer inside the module.

**Uniform Template**. The most immediate distribution to evaluate is, as the original Neocognitron, an uniform distribution of filters. Computing the number of filters in an uniform distribution is straightforward. Adding up the filters in each layer from the base model and divide them by the number of layers give the number of filters to be set in each of them. Formally, we compute the new number in each layer as $f'_l = F/D \quad \forall l \in \{1, ..., D\}$.

In this way, changing the distribution for a VGG19 model built exclusively with sixteen convolutional layers, one final unchangeable fully connected layer and a total number of filters of $F = 5504$ produces a model with $f'_l = 5504/16 = 344$ filters in each layer.

**Reverse Template**. Another straight-forward transformation for the filter distribution adopted in this paper is reversing the number of filters in every layer. Our final model with this template is defined by the filters $f'_l = f_{D-l+1}$.

**Quadratic Template**. The third distribution we evaluated is characterised by a quadratic equation $f'_l = al^2 + bl + c$ and consequently, has a parabolic shape with the vertex in the middle layer. We set this layer to the minimal number of filters in the base model $f_{min} = min(f_l) \quad l \in \{1, ..., D\}$ so, the number of filters is described by $f'_{D/2} = f_{min}$. Also, we find the maximum value in both the initial and final convolutional layers , thus $f'_1 = f'_D$.

To compute the new number of filters in each layer we solve the system of linear equations given by *i)* the restriction of the total number of filters in $\sum_{l=1}^{D} (f'_l) = \sum_{l=1}^{D} (al^2 + bl + c) = F$, that can be reduced to $\left(\frac{D^3}{3} + \frac{D^2}{2} + \frac{D}{6}\right) a + \left(\frac{D^2}{2} + \frac{D}{2}\right) b + Dc = F$, *ii)* the equation produced by the value in the vertex $f'_{D/2} = \frac{D}{2}^2 a + \frac{D}{2} b + c = f_{min}$ and *iii)* the equality from the maximum values which reduces to $(D^2 - 1)a + (D - 1)b = 0$.

**Negative Quadratic Template**. The final template is also a parabola but with the vertex in a maximum, that is, a negative quadratic curve. The equation is the same quadratic equation that the previous template but the restrictions change. Instead of defining a value in the vertex, $f'_l$ at the initial and final convolutional layers are set to the minimal number of filters in the base model $f'_l = f_{min} \quad l \in \{1, D\}$. The number of filters in each layer is computed again with a system of equations specified by *i)* the restriction of the total number of filters as in the quadratic template, and the two points already known in the first and last convolutional layers defined by *ii)* $a + b + c = f_{min}$ and *iii)* $D^2 a + Db + c = f_{min}$.

Once the model has been readjusted with the new number of filters per layer we use a *width multiplier* to test different levels of model compression to make comparable evaluations given that the change in the distributions of kernels modifies the number of parameters, memory consumption and speed. The width multiplier only reduces or increases the set of new filters $f'_l$ proportionally in every layer.

DATASETS

We trained over two datasets traditionally used for convolutional network evaluation: CIFAR-10 and CIFAR-100 (Krizhevsky et al., 2009). Both datasets contain a train set of 50,000 images and a test set of 10,000 images with a resolution of 32x32 and three colour channels. They were published for classification tasks for ten and one hundred classes respectively. Some models were tested also

in Tiny-Imagenet dataset which is a reduced version of the original Imagenet dataset with only 200 classes and images with a resolution of 64 x 64 pixels.

## CONVOLUTIONAL NETWORK MODELS

The state-of-the-art networks evaluated represent some of the highest performing CNNs on the ImageNet challenge in the previous years (Russakovsky et al., 2015). They have been primarily tested on classification tasks and also have demonstrated a strong ability to generalise to images outside the ImageNet dataset. Therefore, it is expected they perform well in the CIFAR datasets.

The VGG network architecture (Simonyan & Zisserman, 2014) is recognised by its simplicity. It is composed by sequential convolutional layers followed by max-pooling reduction layers. The final classification is managed by fully-connected layer and a Softmax classifier. The main disadvantage of these networks is the size of their parameters. In theses paper we use the version of the model with just one fully connected layer in the final classification section.

ResNet (He et al., 2016) succeeds on the problem of training very deep CNNs by reformulating the assumption that the network blocks are modelling a function closer to an identity mapping than to a zero mapping. Therefore, it should be easier to find differences with reference to an identity rather than to a zero mapping. This assumption is carry out by adding additional references at the end of building blocks.

The Inception/GoogleNet architecture (Szegedy et al., 2015) make use of the Inception module conceived as a multi-level feature extractor allowing simultaneous extraction features of several sizes within the same module of the network.

The MobileNet network (Howard et al., 2017) is built on depthwise separable convolutions except for the first layer which is a full convolution. All layers are followed by a batch normalisation and ReLU nonlinearity with the exception of the final fully connected layer which consist of a softmax layer for classification.

## 4    MODELS COMPARISON UNDER SIZE, MEMORY FOOTPRINT AND SPEED

In this section we first investigate the effects of applying different templates to the global distribution of kernels in well known convolutional neural network models (VGG, ResNet, Inception and MobileNet). We compare models under the basis of size, memory and speed in two of the popular datasets for classification tasks.

All experiments have models fed with images with the common augmentation techniques of padding, random cropping and horizontal flipping. Our experiments were run in a NVidia Titan X Pascal 12GB GPU adjusting the batch size to 16 samples to fit in GPU Memory at the maximum model scaling.

## TEMPLATE EFFECT OVER THE BASELINE MODELS

We conducted a first experiment to test our proposed templates on the selected architectures. All convolutional models, with and without templates, were trained for 160 epochs using the same conditions: stochastic gradient descent (SGD) with a scheduled learning rate starting in 0.1 for the first 80 epochs, 0.01 for the next 40 epochs and finally 0.001 for the remaining epochs.

The results are presented in Table 1. It is shown that for VGG, ResNet and MobileNet, the model accuracy improves in CIFAR-10 when the templates are applied. Inception architecture presents the highest accuracy of all base models in both datasets and templates are only able to change its accuracy in less than 1.5%. This is surprising given the drastic modifications that the model is suffering after the change of filter distribution. Models that share a sequential classical architecture such as VGG and MobileNet, show a better improvement when using templates. We additionally performed an experiment with some of the models on the Tiny-Imagenet dataset (See Table 2). While there is a very modest reduction in parameters with VGG, a very remarkable accuracy improvement is produced in MobileNet.

| Dataset | Model | Base | Redistribution Templates | | | |
| --- | --- | --- | --- | --- | --- | --- |
| | | | Reverse-Base | Uniform | Quadratic | Negative-Quadratic |
| CIFAR-10 | VGG-19 | 93.52 | **94.40** | 94.24 | 94.18 | 94.21 |
| | ResNet-50 | 94.70 | 95.17 | 95.08 | 94.41 | **95.23** |
| | Inception | 94.84 | 94.60 | 94.82 | **94.86** | 94.77 |
| | MobileNet | 89.52 | **91.35** | 91.28 | 89.98 | 91.04 |
| CIFAR-100 | VGG-19 | 71.92 | **74.65** | 74.03 | 73.55 | 74.05 |
| | ResNet-50 | 77.09 | 74.80 | 76.65 | 75.71 | 76.76 |
| | Inception | 78.03 | 77.78 | **78.12** | 77.67 | 76.65 |
| | MobileNet | 65.08 | 66.39 | **68.71** | 63.89 | 67.05 |

Table 1: Model performances with the original distribution and four templates for the same number of filters evaluated on CIFAR-10 and CIFAR-100 datasets. After filter redistribution models surpass the base accuracy. Results show average of three repetitions.

| Model | Base | Redistribution Templates | | | |
| --- | --- | --- | --- | --- | --- |
| | | Reverse-Base | Uniform | Quadratic | Negative-Quadratic |
| VGG-19 | **97.23** / 25.0 | 83.67 / 20.6 | 95.37 / **19.3** | 96.25 / 20.7 | 89.45 / 20.6 |
| MobileNet | 58.52 / 3.4 | 63.90 / **2.4** | 69.45 / **2.4** | **77.97** / 3.3 | 66.21 / 2.6 |

Table 2: Model performances (left) and parameters (right) with the original distribution and four templates for the same number of filters evaluated on Tiny-ImageNet dataset. After applying templates, models improve in accuracy or resource utilisation.

| | Model | Base | Redistribution Templates | | | |
| --- | --- | --- | --- | --- | --- | --- |
| | | | Reverse-Base | Uniform | Quadratic | Negative-Quadratic |
| Parameters (Millions) | VGG-19 | 20.0 | 20.0 | 16.0 | **15.8** | 20.0 |
| | ResNet-50 | 23.5 | 23.1 | **12.9** | 19.0 | 33.0 |
| | Inception | **6.2** | 6.7 | **6.2** | 7.2 | 7.0 |
| | MobileNet | 3.2 | **2.2** | **2.2** | 3.2 | 2.4 |
| Memory Footprint (GB/batch) | VGG-19 | **1.3** | 2.6 | 4.4 | 2.0 | 1.4 |
| | ResNet-50 | 3.1 | 11.5 | 4.1 | 7.9 | **3.0** |
| | Inception | **1.5** | 3.1 | 1.7 | 2.2 | 1.6 |
| | MobileNet | 2.5 | 5.1 | 1.5 | 6.0 | **1.0** |
| Inference Time (ms/batch) | VGG-19 | **3.0** | 8.2 | 5.3 | 7.5 | 7.3 |
| | ResNet-50 | 46.4 | 61.0 | **23.4** | 59.0 | 47.6 |
| | Inception | 28.5 | 54.9 | 34.3 | 25.2 | **24.3** |
| | MobileNet | **3.8** | 6.8 | 4.3 | 7.4 | 4.9 |

Table 3: Parameters, memory and inference time for selected models when applying our templates keeping the same number of filters evaluated on the CIFAR-10 dataset. Models are normally optimised to exploit GPU operation, therefore the original base distribution has a good effect in speed but the redistribution of filters induced by our templates makes models capabilities improve on the other metrics. Memory footprint is shown as reported by CUDA.

When evaluated under other metrics (Table 3), models are affected differently with each template and model. The Reverse-Base, Uniform and Quadratic templates show some reductions in the number of parameters while Negative Quadratic template reduces the memory usage. Inference Time is affected negatively for the templates. This is an expected result as original models are designed to perform well in the GPU. The Inception model shows an improvement in speed with a reduction of 14% over inference time respect to the base model. It is important to notice that a reduced number of parameters does not correspond to a low consumption of memory, not even a small inference time. Some of the causes are the difference in feature map resolution for filters in different layers, the need to keep early feature maps in memory for late layers and the restrictions for improving parallelisation in the computational graph of the model.

TEMPLATE EFFECT WITH SIMILAR RESOURCES

It can be argued that models obtained with templates make use of more resources such as memory or number of operations in the GPU (reflected in the low inference speed). So, we formulated a second experiment that makes proportional changes in the models after applying the templates. We not only apply reductions to the models but also increments in order to observe if the actual total number of filters is adequate for the task the model is performing or if the model accuracy could improve by adding more filters. Thus, we create curves for each template applying an uniform reduction using a width multiplier with values of 1,6, 1,3, 1.0, 0.8, 0.5, 0.25, 0.1 and 0.05. These curves of reduction allow comparison under the same amount of resources as well as compares the use of resources under the same accuracy. The experiment also shows the level of reduction that our models can tolerate without a significant loss in accuracy.

We add dashed lines to every plot to be used as a reference for the model with the original distribution and no reductions which is the point where both vertical and horizontal dashed lines cross. In general, any arbitrary vertical line in the plot compares accuracy between models with amount of resources (parameters, memory or speed). On the other side, any arbitrary horizontal line compares the resources taken for each model under each template to produce similar accuracy.

Evaluating a model performance using accuracy and parameters is by far the default approach. We show models performances with these metrics in figure 2. VGG and MobileNet models improve in accuracy almost with any template in CIFAR-10 and CIFAR-100. Under reductions, their original accuracy can be reached with less than 25% of the original parameters in the two models. ResNet shows less improvement when compared with networks with similar resources, yet templates can reduce the model further before accuracy drops. Inception behavior considering the same resources remains similar no matter the template used. In general, for this test, the uniform template seems to get the best parameter efficiency for all the models.

We are convinced that for practical implementations, comparing parameters is not a good option. Table 3 has shown that models with a small number of parameters are not necessary related with a small memory footprint or bigger speed and more results are presented in Figure 3 in the Appendix. We observe again that VGG and MobileNet accuracy is enhanced by templates. More than 50% of memory consumption can be reduced in both models while producing the same accuracy. With this metric, ResNet and Inception improve slightly in CIFAR-10 with the Negative-Quadratic template but they perform lower with the rest of templates. We attribute the lower efficiency in memory to the fact that in all the templates but Negative-Quadratic, the number of filters is increased in the initial layers. At these layers the size of feature maps produced for each filter is bigger, and therefore, more memory costly.

One final comparison also important for practical issues is inference time. Our experiments show the patter of improvement for VGG and MobileNet and a degradation of inference time when adopting templates in ResNet and Inception (See Figure 4 in Appendix). In particular Inception shows an improvement with the Negative-Quadratic template in CIFAR-10.

By looking results in inference time it can look unpromising to apply templates. However we can take a different perspective, by sacrificing inference speed it is possible to obtain models with a better accuracy. This could be an unwanted decision but it is frequently taken. It is clearly stated in the inference time between different original models. For example by using ResNet the accuracy improves compared to the obtained by VGG in the two datasets tested, but at the cost of increasing the time for inference. On the contrary, looking for speed enhancement MobileNet has sacrificed

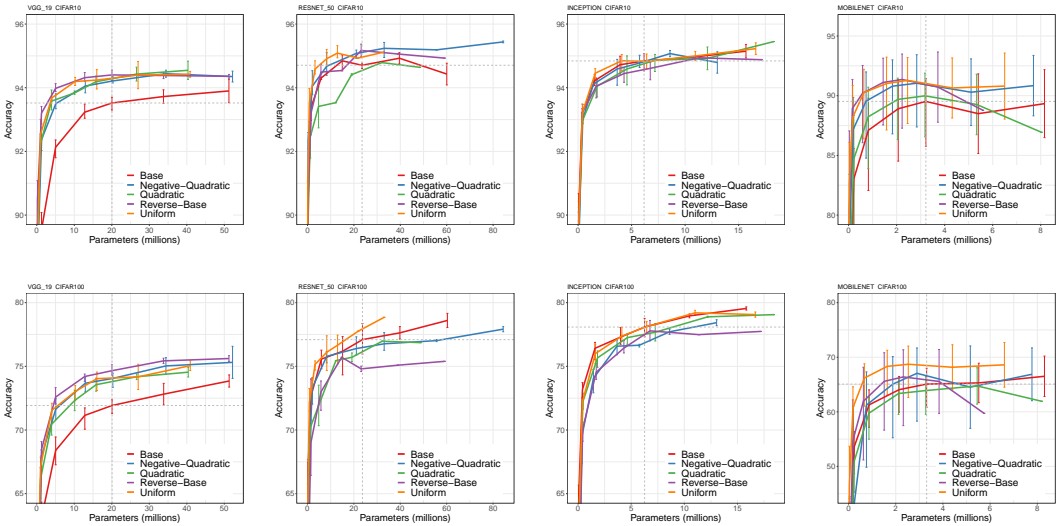

Figure 2: Average Accuracy versus Parameters in CIFAR-10 (top row) and CIFAR-100 (bottom row) datasets using templates with VGG, ResNet, Inception and MobileNet. Curves are created by reducing models using width multiplier scaling. An arbitrary vertical line in the plot compares templates effects using the same amount of parameters. An arbitrary horizontal line compares parameters for reaching the same accuracy. Errors bars denote maximum and minimum of three repetitions.

accuracy. In these sense, our templates are still competitive when compared to searching for a totally different model in order to improve accuracy.

## 5 CONCLUSIONS

The most common design of convolutional neural networks when choosing the distribution of the number of filters is to start with a few and then to increase the number in deeper layers. We challenged this design by evaluating some architectures with a varied set of distributions on the CIFAR and TinyImagenet datasets. Our results suggest that this pyramidal distribution of filters is not necessarily the best option for obtaining the highest accuracy or even the highest parameter efficiency.

The method presented allows a model architect to apply a set of templates for redistributing the number of filters originally assigned to each layer in existent convolutional network models before training the models from scratch. This redesign and the following proportional reduction can be achieved without any previous training process to select particular weights. In essence, the application of filter redistribution templates offers an alternative and or additional approach to the iteration-intensive architecture search or model pruning.

Our experiments show that the models, with the same amount of filters but a different distribution produced by our templates, improves accuracy with up to 5.5% improvement for some model-task pairs. But after being pruned uniformly, they can obtain the same accuracy than the original models using less resources such as number of parameters with up to 45% less parameters and a memory footprint up to 60% smaller.

Results also reveal an interesting behaviour in the evaluated models: a strong resilience to changes in filter distribution. The variation in accuracy for all models after administering templates is less than 5% despite the modifications in the distributions, and therefore in the original design, are considerable. This finding strengthen our belief that it is not worth exploring the whole space of filter distributions to find the best solution at the cost of training models for a large number of iterations, as this solution possibly won't be too distant of the baseline performance. However, it is possible to explore just some distinct and easy to implement distributions such as those represented by our templates, that produces benefits depending on the resource to be optimised.

Our work overall offers an additional tool to model designers, both automated and manual, and we hope it motivates further work for iteration-less methods and help gather data to build understanding of the design process for model-task pairs.

ACKNOWLEDGMENTS

Removed for blind review

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

APPENDIX

5.1   EXPERIMENTS COMPARING MODELS MEMORY FOOTPRINT AND SPEED

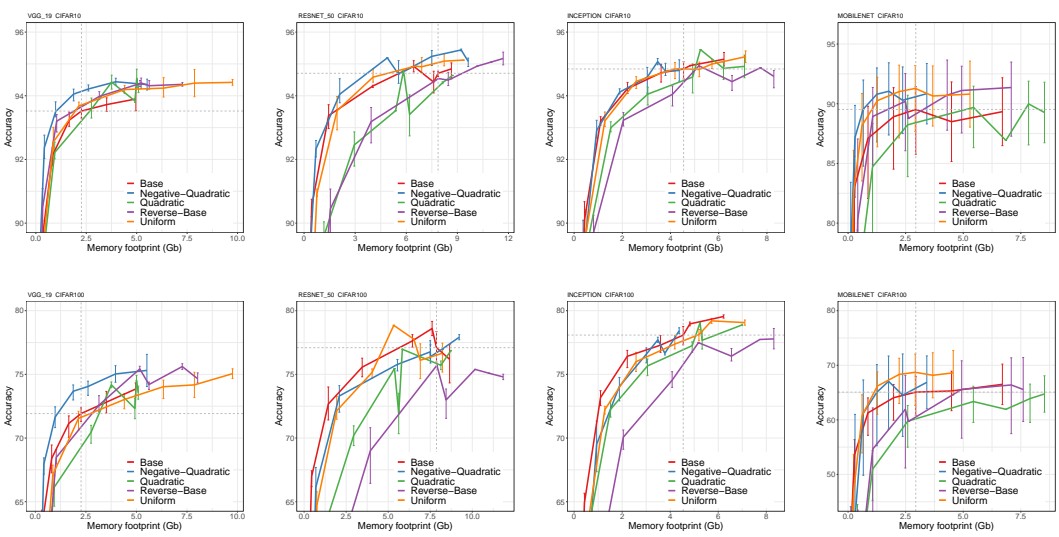

Figure 3: Accuracy versus Memory Footprint (batch size = 64) in CIFAR-10 (top row) and CIFAR-100 (bottom row) datasets using templates with VGG, ResNet, Inception and MobileNet. Curves are created by reducing models using width multiplier scaling. An arbitrary vertical line in the plot compares templates effects in accuracy using the same amount of memory. An arbitrary horizontal line compares memory consumption for reaching the same accuracy. Dashed lines crosses in the original model with no reduction.

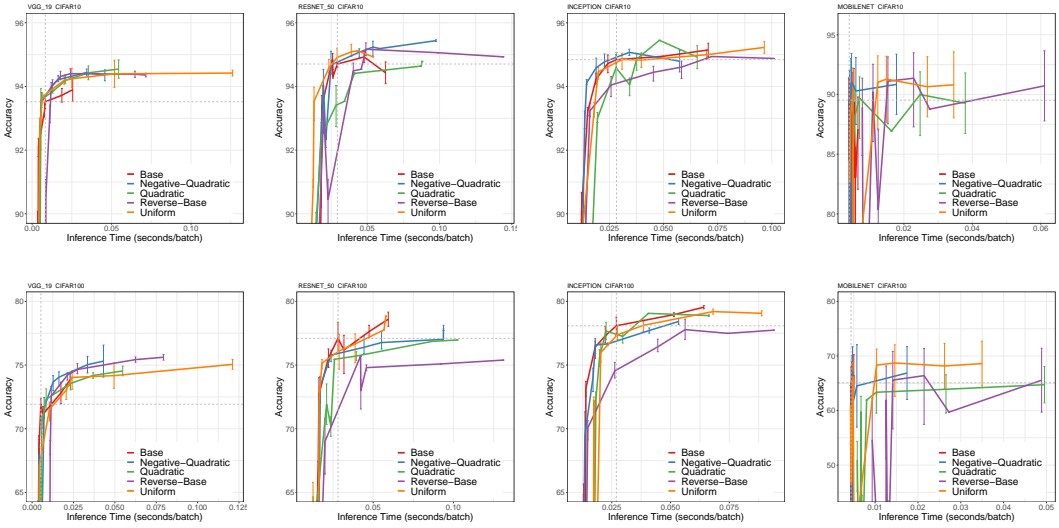

Figure 4: Accuracy versus Inference Time (batch size = 64) in CIFAR-10 (top row) and CIFAR-100 (bottom row) datasets using templates with VGG, ResNet, Inception and MobileNet. Curves are created by reducing models using width multiplier scaling. An arbitrary vertical line in the plot compares templates effects in accuracy between models with same inference speed. An arbitrary horizontal line compares inference time of models with the same accuracy.

## 5.2 FILTER DISTRIBUTION FOR EXISTING CONVOLUTIONAL DEEP NETWORK MODELS

During the design of a convolutional deep network architecture the number of filters, required at every layer, is selected. Neocognitron model keeps the same number of filters across all layers in the model. A distribution of filter widely used forms a by-pyramidal architecture. The number of filters across the different layers is usually increased as the size of the feature maps decrease. We present in Figure 5 the distribution for the models VGG, ResNet, Inception and MobileNet which were tested in our experiments.

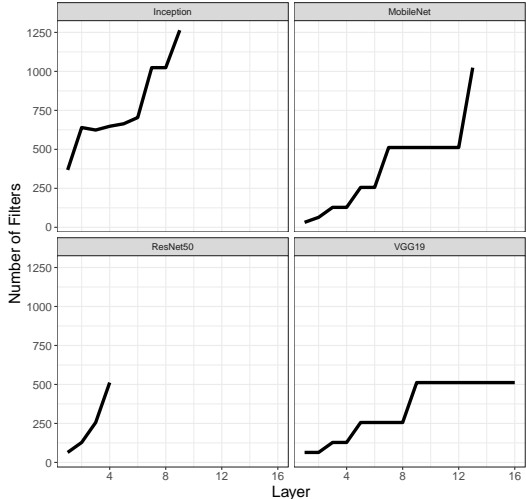

Figure 5: Increasing filters per layer is a common design in convolutional models. We apply templates to existing models to change filters distribution. (ResNet shows filters per sets of residual blocks. The number is keep fixed inside them).

