# OpenReview forum: "Filter redistribution templates for iteration-lessconvolutional model reduction"
_ICLR.cc/2020/Conference — Reject_

### Official Review · AnonReviewer2 · 2019-10-05
**Official Blind Review #2**

**Rating:** 6

**Review:**

The search/design space of neural network architectures is vast, so researchers tend to use simple heuristics to guide their designs; alternatively neural architecture search methods may minimise heuristics in order to remove bias within the search. The authors propose applying a few simple heuristics in the form of "templates" for the number of convolutional filters across the different layers within an architecture. Apart from reversing the filter distribution, which starts a network with a large amount of filters and then reduces them (given how the filter number is generally increased with depth and reduction in spatial resolution), the authors also propose using the same number of filters per layer ("uniform"), as well as "quadratic" and "negative quadratic" distributions. There does not seem to be any particular motivation for these patterns, but this is at least better than poor justifications.

The results include 4 common CNN architectures and experiments on CIFAR-10 and CIFAR-100, which is acceptable, but only single results are shown, which makes it hard to judge the significance of the performance of the redistributed networks, especially given that there seems to be no particular trend in the performance of any of the templates. The accuracy vs. parameter count results are interesting, as the default architecture does worse a good amount of the time. Unfortunately memory footprint results are mixed, and the default architectures tend to perform much better with respect to inference time. Most of the figures are hard to read and not labelled completely (e.g. relying on the caption to dictate which column corresponds to which architecture), so these should be reworked.

Ultimately, the results are largely empirical, and the work would benefit from a better exploration for the consequences of using these templates, and any sort of rule or correlation that links these to successful architectures - at the moment it is only a somewhat interesting observation. Combined with the lack of multiple runs to provide more solid evidence for the empirical findings, I would reject this work.

**Experience Assessment:**

I have read many papers in this area.

**Review Assessment: Checking Correctness Of Derivations And Theory:**

N/A

**Review Assessment: Checking Correctness Of Experiments:**

I carefully checked the experiments.

**Review Assessment: Thoroughness In Paper Reading:**

I read the paper thoroughly.

---

> ### Author Response · Authors · 2019-11-15
> **Re: Official Blind Review #2**
>
> We are aware that, while our templates are heuristically chosen, they open avenues for insight for designing models and further work on finding what other templates to be used. Our aim is to show the benefits that a one-shot redistribution of filters in a model could produce. The architecture-task pair is likely to be unique with respect to the best template that works for it. While this may sound intractable, the small number of templates here proposed already offer a fast and iteration-less alternative to architecture search or optimisation which can deliver better performance straight away.
>
> In sections 1 and 3 we expanded on the reasons to choose some of our templates. But to clarify, the first is the original Neocognitron architecture (uniform template) and the second is the patterns observed from some papers, particularly MorphNet where at least three behaviours are present in different blocks from ResNet101 model: 1) filters increase in deeper layers, 2) filters agglomerate in the centre and 3) filters are reduced in the centre of the block. Smallify shows also pattern number 2 for a vgg model.
> We already clarified this motivation and inspiration from these sources.
>
> We have now included error bars, changed the color palette and provide labels for models and datasets in each plot.

---

> > ### Comment · AnonReviewer2 · 2019-11-15
> > **Issues Addressed**
> >
> > References to prior work and architectures now give a better motivation and background to this work. Having results averaged over multiple runs now gives me more confidence in the results, but these should be included in the tables, not just the plots. For the plots, shaded error envelopes would make determining the overlap in performance easier to distinguish (as compared to error bars) - particularly with MobileNet. The inclusion of TinyImagenet also improves the empirical results - extending results to include the other configurations should be done for the final version though. As the authors have largely addressed my concerns I will update to accept.

---

### Official Review · AnonReviewer3 · 2019-10-23
**Official Blind Review #3**

**Rating:** 3

**Review:**

This paper changes the distribution of number of filters (called “filter distribution template”, or "template") at each layer in modern deep Conv models (e.g., VGG, Inception, ResNet) and discover that the model with unconventional (e.g., reverse base, quadratic) template sometimes outperform conventional one (when f_{l+1} = 2 f_l).

One big issue of this paper is that it didn't mention any theoretical reason why the total number of filters is the decisive factor for the test performance. It is not justified at all and the empirical result is also mixed (See Table 1). This brings about the question mark of the motivation of this paper from the first place. In contrast, empirically people have observed that a model with more parameters (and/or more FLOPs) within the same architecture family gives better performance. This is not directly related to the total number of filters, which the main topic in the paper.

As a result, it is not clear whether a gain of the performance is simply due to the change of #parameters/FLOPs or due to the fact that different distribution templates are used. As shown in Table. 2, there is huge variation in terms of #parameters and FLOPs between different versions of the same network, making the comparison fairly difficult and inconclusive. I would strongly suggest the authors to compare the performance between different templates when keeping #parameters and/or FLOPs fixed. This should be easy to do by computing how many filters are needed per layer to reach the desired #parameters/FLOPs, while keeping the desired distribution.

Also, to make a strong conclusion, they paper should also report ImageNet results trained with different templates.

Overall, the paper, in its current form, is not ready for publication and I would vote for rejection.

=========Post Rebuttal=======
I reread the paper after authors revision and rebuttal. Thanks authors for the hard work.

Indeed the authors have compared different templates when the number of parameters remain approximately the same (in both the original version and the new revision). I overlooked it and apologize.

However, after rereading the paper, the conclusion is still not that clear and I didn't see a clear take home message about which filter template is better than the other. In the section "Template Effects with similar resources", it seems that uniform template patterns is the best for many models, which somehow is negative results given the motivation of this paper.

In addition, when comparing Fig. 3 in the original version versus that (Fig. 3) in the revision, some curves have changed their shape drastically (e.g., MobileNet on CIFAR10 and CIFAR100) and uniform template shows stronger dominance. This worries me a bit that the experiments might still be preliminary and the paper is yet not ready for publication.

 I will keep the score.


**Experience Assessment:**

I have published one or two papers in this area.

**Review Assessment: Checking Correctness Of Derivations And Theory:**

I assessed the sensibility of the derivations and theory.

**Review Assessment: Checking Correctness Of Experiments:**

I assessed the sensibility of the experiments.

**Review Assessment: Thoroughness In Paper Reading:**

I read the paper at least twice and used my best judgement in assessing the paper.

---

> ### Author Response · Authors · 2019-11-15
> **Re: Official Blind Review #3**
>
>
> In section “Template effect with similar resources” we provide several experiments consisting in changing proportionally the width of every model after applying a template, in order to increase and decrease its parameters/FLOPs.
>
> Our results are consistent with the idea that a model with more parameters (and/or more FLOPs) within the same architecture family gives better performance but what we also show is that using the same parameters with two different templates in the same model delivers different accuracy.
>
> As per R4 suggestion we include results for TinyImagenet that strengthen the approach’s applicability.
>
> We have thoroughly changed description and wording.

---

### Official Review · AnonReviewer1 · 2019-10-29
**Official Blind Review #1**

**Rating:** 3

**Review:**

This paper investigates the impact of several predefined filter templates, including uniform template, reverse template, quadratic template, and negative quadratic template, to the performance of different neural networks such as VGG-19, ResNet-50, Inception Network, and MobileNet.

This paper uses some templates. However, it is impossible to enumerate all possible templates and hence some good templates may not be included and studied in this paper, which makes the empirical studies in this paper less useful.

In experiments, authors need to compare with the results of neural architecture search. Based on such comparison, we can see whether the templates used in this paper are reasonable.

In experiments, it seems that different templates may have their own characteristics and their usefulness also depends on the neural network used. So it is not easy to make general conclusion.

**Experience Assessment:**

I have read many papers in this area.

**Review Assessment: Checking Correctness Of Derivations And Theory:**

I assessed the sensibility of the derivations and theory.

**Review Assessment: Checking Correctness Of Experiments:**

I assessed the sensibility of the experiments.

**Review Assessment: Thoroughness In Paper Reading:**

I read the paper at least twice and used my best judgement in assessing the paper.

---

> ### Author Response · Authors · 2019-11-15
> **Re: Official Blind Review #1**
>
> As R1 points out, exploring over all possible templates is intractable, even when many methods (“Neural Architecture Search: A Survey”) explore a reduced space, the amount of computational resources is highly costly. Our assumption is that there exists a small set of filter distributions that we called templates, that are distinct and easy to apply and able to promote improvements to existing deep network models. We have extended this explanation about the justification of using templates in section 3.
>
> In figure 1, we show the distribution of filters for the four architectures tested in the experiments. All of them follow the same pattern of increasing filters in deeper layers. We highlight that there is no universal distribution of filters that works well for every model-task pair. By using the proposed templates we offer a fast and iteration-less alternative to architecture search which can deliver better performance straight away.

---

### Official Review · AnonReviewer4 · 2019-10-30
**Official Blind Review #4**

**Rating:** 6

**Review:**

This paper presents a simple methodological study on the effect of the distribution of  convolutional filters on the accuracy of deep convolutional networks on the CIFAR 10 and CIFAR 100 data sets. There are five different kind of distributions studied: constant number of filters, monotonically increasing and decreasing number of filters and convex/concave with a local extremum at the layer in the middle.  For these distributions, the total number of filters is varied to study the trade-off between running-time vs. accuracy, memory vs. accuracy and parameter count vs. accuracy.

Although the paper is purely experimental without any particular theoretical considerations, it presents a few surprising observations defying conventional wisdom:
- The standard method of increasing the number of filters as the number of convolutional nodes is increasing is not the most optimal strategy in most cases.
- The optimal distribution of channels is highly dependent on the network architecture.
- Some network architectures are highly stable with respect to the distribution of channels, while others are very sensitive.

Given that this paper is easy to read and presents interesting insights for the design of convolutional network architectures and challenges mainstream views, I would consider it to be a generally valuable contribution, at least I enjoyed reading it.

Despite the intriguing nature of this paper, there are several weaknesses which make me less enthusiastic about the quality of the paper:
- The experiments are done only on CIFAR-10 and CIFAR-100. These benchmarks are somewhat special. It would be useful to see whether the results also hold for more realistic vision benchmarks. Even if running all the experiments would be costly, I think that at least a small selection should be reproduced on OpenImages or MS-Coco or other more realistic benchmarks to validate the findings of this paper.
- It would be interesting to see whether starting from the best channel distributions, applying MorphNet would end up with different distributions. In general: whether MorphNet would end up with similar distributions automatically.
-  The paper does not clarify how the channel sizes for Inception were distributed, since proper balancing of the 1x1 and more spread out convolutions is a key part of that architecture. This is not clarified in this paper.
- The grammar of the paper is poor, even the abstract is hard to read and interpret.
- The paper presents itself as a methodology for automatically generating the optimal number of channels, while it is more of a one-off experiment and observation than a general purpose method.

Another small technical detail regarding the choice of colors in the diagrams: the baseline distribution and constant distribution are very hard to distinguish. This is especially critical because these are the two best distributions on average. Also the diagrams could benefit from more detailed captions.

The paper presents interesting, valuable experimental findings, but it is not extremely exciting theoretically. Also its practical execution is somewhat lacking. If it contained at least partial results on more realistic data sets, I would vote for strong accept, but in its current form, I find it borderline acceptance-worthy.


**Experience Assessment:**

I have published in this field for several years.

**Review Assessment: Checking Correctness Of Derivations And Theory:**

N/A

**Review Assessment: Checking Correctness Of Experiments:**

I assessed the sensibility of the experiments.

**Review Assessment: Thoroughness In Paper Reading:**

I read the paper thoroughly.

---

> ### Author Response · Authors · 2019-11-15
> **Re: Official Blind Review #4**
>
>
> We provide additional comparative results in Tables 1 and 2 for tiny-imagenet dataset (thank you for the suggestion) which is a 200 classes subset from Imagenet. The results confirm some models benefit by redistributing their filters with our templates.
>
> Our aim is to provide an insight about the benefits that a redistribution of filters in a model could produce but further exploration should be done. The architecture-task pair is likely to be unique with respect to the best template that works for it. While this may sound intractable, the small number of templates here proposed already offer a fast and iteration-less alternative to architecture search or optimisation which can deliver better performance straight away.
>
> In sections 1 and 3 we mention two of the reasons to choose some of our templates. But to clarify, the first if the original Neocognitron architecture (uniform template) and the second is the patterns observed from some papers, particularly MorphNet where at least three behaviours are present in different blocks from ResNet101 model: 1) filters increase in deeper layers, 2) filters agglomerate in the centre and 3) filters are reduced in the centre of the block. Smallify shows also pattern number 2 for a vgg model.
> We already clarified this motivation and inspiration from these sources.
>
> Regarding the number of filters within Inception modules, we changed the distribution of channels by keeping a constant number of filters for each filter size, except Reverse-Base template which have the original distribution inside each module but assigned inversely, that is the first module has the distribution of the last one and vice versa. This was an unexpected behavior as we are disturbing the model design in a key aspect, as the reviewer mentioned, and still the model performance remains similar in some datasets.
> We already clarified the existing description of how we distributed channel sizes in section 3.
>
>
> We present our work as both insight and as a methodology in the sense that an improvement in the performance of an existing architecture can be made by following a set of straight-forward steps: take a base mode, apply the template and reduce the model proportionally according to a budget restriction before training the model again. Our main aim is to highlight the importance of filter distributions, provide an additional design methodology and motivate further exploration in this space.
>
>
> We have changed the color palette and provide labels for models and datasets in each plot.

---

### Author Response · Authors · 2019-11-15
**Re: All reviewers**

We thank our reviewers for their valuable comments. We have thoroughly improved the document based on your guidance. We believe we have addressed all the issues raised in our response below, including the addition of further evaluation to demonstrate the applicability and generalisation of our approach. We address each issue individually.

---

### Decision · Program_Chairs · 2019-12-19

**Decision:**

Reject

**Comment:**

This paper examines how different distributions of the layer-wise number of CNN filters, as partitioned into a set of fixed templates, impacts the performance of various baseline deep architectures.  Testing is conducting from the viewpoint of balancing accuracy with various resource metrics such as number of parameters, memory footprint, etc.

In the end, reviewer scores were partitioned as two accepts and two rejects.  However, the actual comments indicate that both nominal accept reviewers expressed borderline opinions regarding this work (e.g., one preferred a score of 4 or 5 if available, while the other explicitly stated that the paper was borderline acceptance-worthy).  Consequently in aggregate there was no strong support for acceptance and non-dismissable sentiment towards rejection.

For example, consistent with reviewer comments, a primary concern with this paper is that the novelty and technical contribution is rather limited, and hence, to warrant acceptance the empirical component should be especially compelling.  However, all the experiments are limited to cifar10/cifar100 data, with the exception of a couple extra tests on tiny ImageNet added after the rebuttal.  But these latter experiments are not so convincing since the base architecture has the best accuracy on VGG, and only on a single MobileNet test do we actually see clear-cut improvement.  Moreover, these new results appear to be based on just a single trial per data set (this important detail is unclear), and judging from Figure 2 of the revision, MobileNet results on cifar data can have very high variance blurring the distinction between methods.  It is therefore hard to draw firm conclusions at this point, and these two additional tiny ImageNet tests notwithstanding, we don't really know how to differentiate phenomena that are intrinsic to cifar data from other potentially relevant factors.

Overall then, my view is that far more testing with different data types is warranted to strengthen the conclusions of this paper and compensate for the modest technical contribution.  Note also that training with all of these different filter templates is likely no less computationally expensive than some state-of-the-art pruning or related compression methods, and therefore it would be worth comparing head-to-head with such approaches.  This is especially true given that in many scenarios, test-time computational resources are more critical than marginal differences in training time, etc.